# Single-Blind Randomized Controlled Trial: Comparative Efficacy of Dark Chocolate, Coconut Water, and Ibuprofen in Managing Primary Dysmenorrhea

**DOI:** 10.3390/ijerph20166619

**Published:** 2023-08-21

**Authors:** Kaifar Nuha, Kusnandi Rusmil, Ahmad Rizal Ganiem, Wiryawan Permadi, Dewi Marhaeni Diah Herawati

**Affiliations:** 1Master of Midwifery Study Program, Faculty of Medicine, Universitas Padjadjaran, Jl. Eyckman No. 38, Bandung 40161, Indonesia; 2Department of Child Health, Faculty of Medicine, Universitas Padjadjaran, Jl. Eyckman No. 38, Bandung 40161, Indonesia; 3Dr. Hasan Sadikin General Hospital, Bandung, 40161, Indonesia; 4Department of Neurology, Faculty of Medicine Universitas Padjadjaran, Jl. Eyckman No. 38, Bandung 40161, Indonesia; 5Department of Obstetric and Gynecology, Faculty of Medicine, Universitas Padjadjaran, Jl. Eyckman No. 38, Bandung 40161, Indonesia; 6Department of Public Health, Faculty of Medicine, Universitas Padjadjaran, Jl. Eyckman No. 38, Bandung 40161, Indonesia

**Keywords:** dysmenorrhea, menstrual pain, ibuprofen, reproductive health, dysmenorrhea management, green coconut water, dark chocolate, magnesium, anti-inflamatory drugs

## Abstract

Dysmenorrhea, the pain experienced by women during menstruation, affects a significant proportion of women worldwide and often leads to decreased productivity. Various pharmacological and non-pharmacological treatments are available for pain relief, but information on their effectiveness, particularly regarding green coconut water, dark chocolate, and Ibuprofen, remains limited. This study aimed to compare the effectiveness of green coconut water, dark chocolate bars, and Ibuprofen in reducing the intensity of primary dysmenorrhea. In this research, a randomized controlled trial with a quantitative design was conducted, involving 45 participants randomly assigned to receive 330 mL of green coconut water, 35 g of 70% dark chocolate, or 400 mg Ibuprofen. The interventions were administered on the first day of menstruation when dysmenorrhea symptoms typically occur in subjects. This study used a single-dose approach to evaluate the immediate impact of each treatment. The subjects were instructed to consume the given interventional product within 15 min. The pain intensity was measured using a Numeric Rating Scale before the intervention and 2 h after the subjects finished consuming the interventional product. The multivariate Kruskal–Wallis test revealed a significant difference in effectiveness among the three interventions (*p* < 0.05). The study found that Ibuprofen was the most effective intervention compared to the other interventions. These findings contribute to understanding the treatment options for primary dysmenorrhea and emphasize the efficacy of Ibuprofen (trial registration: ClinicalTrials.gov: NCT05971186).

## 1. Introduction

Women have a unique reproductive system because they experience menstruation every month. During menstruation, women usually feel various intensities of pain, ranging from mild to severe, called dysmenorrhea [1]. In accordance with its pathophysiology, it is classified as either primary or secondary dysmenorrhea [2]. Primary dysmenorrhea is characterized by spasmodic and painful cramps in the lower abdomen that start before or during menstruation and are not associated with pelvic issues [3]. The onset of primary dysmenorrhea occurs in adolescence, which occurs within 6 to 24 months after the onset of menarche. The pain follows a distinct and cyclical pattern, which usually reaches its peak severity on the first day of menstruation and lasts up to 72 h [4].

The incidence of menstrual pain is relatively high; globally, 50–90% of women of reproductive age experience pain, most of which is caused by primary dysmenorrhea [5]. In Indonesia, specifically, it is reported that 55% of women experience dysmenorrhea during menstruation [6]. Despite the significant prevalence of dysmenorrhea, many women do not report their symptoms, suggesting that the actual prevalence may be even higher than what has been reported [7].

Dysmenorrhea results in a substantial productivity loss [8]. The effects go beyond physical discomfort, influencing women’s capacity to engage in daily tasks and overall well-being [9]. Every month, those women have to experience the torture of menstrual pain, which is why finding the right solution to overcome it is necessary [1]. Dysmenorrhea can be treated by using pharmacological or non-pharmacological management. Pharmacologically, this can be achieved by taking non-steroidal anti-inflammatory drugs (NSAIDs), such as Ibuprofen. Research has shown that by considering the efficacy and safety between Naproxen, Ibuprofen, Diclofenac, Aspirin, and Ketoprofen, Ibuprofen is the recommended optimal over-the-counter analgesic for primary dysmenorrhea [10]. Ibuprofen works by inhibiting the production of prostaglandins, which are hormones that play a pivotal role in the pathomechanism of dysmenorrhea. Non-pharmacologically consuming dark chocolate or green coconut water has also been shown to reduce the intensity of menstrual pain [11,12,13,14].

A previous RCT study conducted in Indonesia specifically compared the efficacy of green coconut water in two different doses, 330 mL and 165 mL, and also with 330 mL mineral water as the control group, showing that the 330 mL green coconut water dose was an effective dose compared to the control. In contrast, another dose had a weak effect. This finding proves that green coconut water can be a non-pharmacological alternative in handling dysmenorrhea pain [15].

In recent years, natural remedies for managing dysmenorrhea have gained attention. Furthermore, green coconut water, dark chocolate, and Ibuprofen are known to work through similar mechanisms in reducing the intensity of menstrual pain. Green coconut water and dark chocolate contain high levels of magnesium, which can help reduce smooth muscle tension and potentially cause vasodilation in blood vessels. The flavonoid content in dark chocolate is also known for its natural anti-inflammatory properties by inhibiting the production of prostaglandin hormones, which are responsible for the pain experienced during menstruation. On the other hand, Ibuprofen is one of the most commonly used nonsteroidal anti-inflammatory drugs (NSAIDs) for the pharmacological management of dysmenorrhea. It works similarly to green coconut water and dark chocolate by inhibiting the production of prostaglandin hormones and causing the vasodilation and relaxation of smooth muscles, aiming to decrease the intensity of menstrual pain [15,16,17].

Therefore, investigating the comparative effectiveness of these interventions, namely green coconut water, dark chocolate, and Ibuprofen, in reducing the intensity of primary dysmenorrhea is essential in determining their potential as alternative management options. This study aims to address this research gap through a randomized controlled trial and provide valuable insights into the effectiveness of these interventions in the context of dysmenorrhea management.

## 2. Materials and Methods

### 2.1. Study Design

This quantitative research study employs the single-blind randomized controlled trial (RCT) design, widely recognized as the gold standard for experimental research to compare different treatments [18]. The RCT design incorporates two vital randomization processes: random sampling and random allocation [19].

To ensure the integrity and reliability of the study, several vital roles are involved in facilitating the research process and guaranteeing the blinding procedure. These roles include the researcher, subjects, independent data analyst, data collector, and assistant. All personnel engaged in this study are kept blind to the information regarding the treatments administered to each subject. Blinding the subjects is not feasible in this study due to the distinct differences in the treatments provided, namely coconut water, dark chocolate bar, and Ibuprofen, which are easily discernible based on their shape, taste, and size. Consequently, the research hypothesis will not be disclosed in the Participant Information Sheet or Informed Consent. Subjects will be instructed not to discuss the interventions received with any involved parties, such as the data collector and research assistant. By implementing this robust research design and ensuring the blinding process, the study aims to generate reliable and unbiased results that contribute to the existing body of scientific knowledge.

### 2.2. Data Collection

This study was conducted at Saleha Midwifery Academy, a higher education institution in Aceh Province, Indonesia. The decision to choose this institution was founded on the outcomes of a preliminary study conducted by the researchers themselves before initiating the current study. This preliminary study aimed to identify an appropriate research site. Based on the data collected from several higher education institutions in Aceh Province, it was determined that the highest incidence of dysmenorrhea occurred at Saleha Midwifery Academy. Subsequently, this institution was selected as the research site for the main study. The preliminary study’s data revealed that out of 95 female students within the academy, 94% experienced dysmenorrhea, with only 8.4% reporting minimal disruption to their daily routines due to menstrual pain.

The data collection period in this study occurred from July to September 2022. Two distinct tools were employed for data collection: a pain intensity observation sheet and a questionnaire. A Numeric Rating Scale (NRS) observation sheet assessed pain intensity. The Numeric Rating Scale (NRS) is recognized as a valid and well-established patient-reported outcome measure, often used to evaluate pain intensity, specifically dysmenorrhea [20]. This scale provides a simple and effective way for individuals to express their pain intensity using numerical values from 0 to 10 [21,22]. The questionnaire gathered demographic information and insights into the characteristics of primary dysmenorrhea experienced by the respondents. It covered various aspects, including demographics, menstrual characteristics, and attributes related to dysmenorrhea. The questionnaire aimed to capture relevant details, such as age, height, weight, body mass index (BMI), age of menarche, length of menstrual cycle, duration of menstruation, initial experience of menstrual pain, commonly used pain management methods, and the impact of pain on daily activities [23].

The researcher meticulously considered subject selection by establishing inclusion and exclusion criteria. The inclusion criteria were as follows: (1) women aged between 17 and 24 years, (2) diagnosed with primary dysmenorrhea based on specific characteristics and assessment, (3) willing to refrain from using any pharmacological or non-pharmacological therapies other than the interventions provided by the researcher, and (4) willing to participate as respondents. The exclusion criteria were: (1) allergies to dark chocolate or young coconut water or contraindications to Ibuprofen consumption, and (2) diagnosed with specific gynecological conditions.

This research conducted a thorough subject selection process, employing specific inclusion and exclusion criteria. The inclusion criteria encompassed: (1) women aged between 17 and 24 years, (2) those diagnosed with primary dysmenorrhea that is characterized by specific criteria, including initial pain onset shortly after menarche, lower pelvic or abdominal pain coinciding with menstrual flow lasting 8–72 h, presence of associated symptoms, like back and thigh pain, headache, diarrhea, nausea, and vomiting, for (3) willingness to abstain from using any treatments apart from the interventions offered in the study and (4) willing respondents. On the other hand, the exclusion criteria consisted of (1) allergies or contraindications to dark chocolate, young coconut water, or Ibuprofen consumption and (2) individuals diagnosed with specific gynecological conditions.

Figure 1 presents a comprehensive visual representation of the study participants’ journey, captured through a meticulously designed flow chart The population in this study was 89 female students who experienced primary dysmenorrhea. The subject for this research comprised 45 female students from Saleha Midwifery Academy who met the inclusion criteria and willingly participated in the study by signing the informed consent form after receiving a detailed explanation. The subject size for this study was calculated using the Federer formula, commonly employed in clinical trials with an RCT design, which resulted in a minimum subject size of 9 subjects per group. However, the researcher decided to include 15 subjects in each treatment group, resulting in a total subject size of 45 subjects due to the presence of three treatment groups.

The study utilized a probability sampling technique, specifically simple random sampling, to randomly select subjects from a numbered list of the accessible population using the random name selection feature in Microsoft Excel. Additionally, random allocation was implemented to assign subjects to either the treatment or control group purely by chance, without any influence from the researcher or consideration of subject preferences. This random allocation was accomplished using a blocking system and a designated application.

Consequently, the subjects were divided into three distinct active comparator groups. The allocation of subjects into each group was randomized using a designated application and a block system. Group 1 received the intervention of young coconut water, Group 2 received the intervention of a 70% dark chocolate bar, and Group 3 received the intervention of 1 tablet of Ibuprofen, a non-steroidal anti-inflammatory drug (NSAID). All the interventions are given within this designated timeframe:Intervention with Ibuprofen: In this active comparator group, participants were administered a single 400 mg Ibuprofen tablet within 15 min. Before consumption and two hours after that, pain intensity assessments were conducted using the Numeric Rating Scale (NRS).Intervention with young coconut water: For this active comparator group, subjects consumed 330 mL of young coconut water within 15 min. Pain intensity evaluations were performed using the Numeric Rating Scale (NRS) before ingestion and two hours subsequently.Intervention with dark chocolate bar: Participants in this active comparator group were given a 35 g dark chocolate bar to consume within 15 min. Pain intensity measurements using the Numeric Rating Scale (NRS) were carried out before and two hours post-consumption.

When the subjects experienced dysmenorrhea during menstruation, they would contact the researcher, and the data collector would provide them with a box labeled with the respondent’s code containing the research tools. The package box included a pain intensity measurement scale, a questionnaire regarding menstrual characteristics, a pen, and the assigned intervention (a 35 g dark chocolate bar, a 330 mL pack of young coconut water, or a 400 mg Ibuprofen tablet), along with an empty envelope labeled with the respondent’s code.

After explaining the purpose of the study and their role in the research process, the data collector accompanied the subjects starting from the pretest and provided further instructions. The data collector obtained written consent from the subjects by having them sign the informed consent form. It is important to note that the data collector remained unaware of the content of each box and remained blinded to the intervention received by each subject until the end of the study.

The data collector would leave the room after ensuring the subjects understood the procedures. The subjects would then consume the assigned intervention based on their treatment group, instructed to be consumed within 15 min. Subsequently, the subjects would complete the menstrual pain intensity sheet two hours after receiving the treatment (post-test). They would then place the completed sheet into the empty envelope labeled with the respondent’s code and return it to the researcher.

The research obtained ethical clearance from the Ethics Committee with reference number 703/UN6.KEP/EC/202; the researcher obtained permission to conduct the study at Saleha Midwifery Academy.

### 2.3. Data Analysis

In this study, the evaluation of data normality emerged as an initial step in the analytical process. This assessment utilized the Shapiro–Wilk normality test, a conventional tool implemented through IBM SPSS Statistics 25. For the Shapiro–Wilk normality test, the significance values (Sig.) for several groups were found to be less than 0.005. This indicates that the data are not normally distributed. The non-normal distribution of the data justified the use of non-parametric tests for subsequent analyses in this study.

The bivariate analysis was conducted using the Wilcoxon test. This test was employed to examine the relationship between the dependent and independent variables, aiming to determine the effectiveness level of each intervention. The Kruskal–Wallis test was performed for the multivariate analysis, followed by post hoc tests. This approach was utilized to assess whether there were significant differences in effectiveness among the various groups.

We investigated the associations between the dependent and independent variables by employing the Wilcoxon test. Its application allowed us to determine the effectiveness levels of each intervention and examine the relationships between variables. Moving on to the multivariate analysis, we employed the Kruskal–Wallis non-parametric statistical test. This test served to assess whether there were significant differences in the effectiveness of the interventions across the groups. Subsequently, post hoc tests were conducted to explore further and compare the specific differences between the groups, helping us gain a comprehensive understanding of the significance of these differences.

The implementation of these statistical tests allowed us to unravel the relationships between variables and determine the effectiveness of each intervention. The Wilcoxon test provided insights into the associations between the dependent and independent variables. In contrast, the Kruskal–Wallis test and post hoc analysis shed light on the variations in effectiveness among the different groups. The study maintained a significance level of *p* < 0.05, and all analyses were caried out using IBM SPSS Statistics 25.

## 3. Results

### 3.1. Distribution of Pain Intensity in the Intervention Group

The characteristics of the women in this study can be seen in Table 1. The total sample size was 45 women, evenly distributed among the treatment groups. The subjects in this study were aged 17–21 years, with the majority being 18 years old, accounting for 14 women (31.1%). Based on the data presented in the table, 91.1% of the women had a normal body mass index (BMI).

Most of the subjects experienced their first menstruation at a normal menarche age, with 37 women (82.2%) falling into this category. The menstrual characteristics of the women can also be observed, with 31 women (68.9%) having a normal menstrual cycle ranging from 28 to 35 days and the majority of the subjects, 33 women (73.3%), having a normal menstrual duration of 2–7 days. All women experienced the onset of dysmenorrhea immediately after menarche, which is a characteristic feature of primary dysmenorrhea. Among the women who experienced menstrual pain, the majority, 36 women (80%), chose to endure the pain without any treatment. The majority of the women, 23 women (51.1%), reported that the pain was still tolerable, while 19 women (42.2%) stated that the pain interfered with some of their daily activities.

These findings provide an overview of the characteristics of the women in the study, including their age distribution, BMI status, age at menarche, menstrual cycle, menstrual duration, and the impact of pain on their daily activities. Statistical analysis using the chi-square test in SPSS was conducted to ascertain potential baseline disparities in basic demographic and clinical attributes among the experimental groups. The results, reflected in the *p*-values exceeding the significance level (<0.05), affirm the absence of statistically significant differences in the abovementioned characteristics across the intervention groups. This finding indicates that the baseline attributes were well-balanced.

### 3.2. The Effect of Young Coconut Water, Dark Chocolate, and Ibuprofen on Primary Dysmenorrhea and Normality Test

Table 2 provides a descriptive analysis of the distribution of pain intensities before and after the intervention in the three treatment groups. Prior to the treatment, the women reported experiencing varying levels of pain. The majority of the women, accounting for 64.4%, reported moderate pain, followed by 31.1% who reported severe pain, and only 4.4% who reported mild pain. In order to assess potential differences in pain intensity among the intervention groups prior to treatment, a chi-square test was employed on the collected data. The resulting *p*-value of 0.411 indicates no significant difference in pain intensity across the intervention groups prior to the initiation of treatments. This suggests that the baseline pain intensity was relatively consistent among the groups before any interventions were administered.

After the intervention, there was a noticeable change in the reported pain intensities. Among the women, 48.9% reported experiencing mild pain, indicating a reduction in pain intensity. Furthermore, 17.8% of the women still reported moderate pain, suggesting some improvement but not a complete alleviation of pain. It is noteworthy that none of the women reported severe pain after the intervention. Additionally, 33.3% of the women reported being pain-free, indicating a significant decrease in pain intensity.

These findings demonstrate the effectiveness of the interventions in reducing pain intensity among women. The results suggest that the treatments, including Ibuprofen, coconut water, and dark chocolate, were successful in providing pain relief, with the majority of women experiencing either mild pain or no pain after the intervention.

Bivariate analysis was conducted to determine the relationship between pain intensity and each of the independent variables. The Wilcoxon test results show that the *p*-values for each intervention group are all less than the significance level of 0.05. This result suggests a significant difference in menstrual pain intensity before and after the intervention. In other words, each intervention, namely dark chocolate 70% 35 g, young coconut water 330 mL, and Ibuprofen 400 mg, reduces menstrual pain intensity.

### 3.3. Comparison of the Effectiveness of Young Coconut Water, Dark Chocolate, and Ibuprofen on Primary Dysmenorrhea

Based on Table 3, the *p*-value from the Kruskal–Wallis Test is less than the significance level of 0.05. This result indicates significant differences in reductions in pain intensity among the intervention groups. The table also shows the Numeric Rating Scale (NRS) results, representing the mean rank of the pain intensity difference before and after the intervention. The highest mean rank of pain intensity reduction is observed in the Ibuprofen 400 mg group, followed by the dark chocolate bar 70% 35 goup and the young coconut water 330 mL group. The difference in mean rank suggests that Ibuprofen is the most effective in reducing menstrual pain intensity among the three interventions. Further post hoc tests were conducted to examine whether there were significant differences between one group and another.

Table 4 shows that the *p*-value between the Ibuprofen 400 mg and young coconut water 330 mL groups indicates a significant difference in reductions in pain intensity between these two groups. Similarly, the *p*-value between the young coconut water 330 mL and dark chocolate bar 70% 35 g groups indicates a significant difference. However, the *p*-value between the dark chocolate bar 70% 35 gand and Ibuprofen 400 mg groups is greater than 0.05. Therefore, there is no significant difference between these two groups.

In summary, the statistical analysis reveals that there are significant differences in the reduction in pain intensity among the intervention groups. Ibuprofen 400 mg is the most effective intervention, followed by dark chocolate 70% 35 g, while young coconut water 330 mL shows a relatively lower effectiveness. The post hoc tests further confirm significant differences between certain groups, indicating varying levels of effectiveness among the interventions.

## 4. Discussion

The current study aimed to investigate the comparative effectiveness of three interventions, namely Ibuprofen, dark chocolate, and young coconut water, in reducing menstrual pain intensity in individuals with primary dysmenorrhea. The findings provide valuable insights into the management options for this common condition.

The results of the study indicate that Ibuprofen, a gold-standard treatment for primary dysmenorrhea, demonstrated greater effectiveness in reducing menstrual pain intensity compared to both dark chocolate (70% cocoa, 35 g) and young coconut water (330 mL). These findings are consistent with previous research by Corson and Bolognese that established the efficacy of Ibuprofen in managing dysmenorrhea. That study mentioned that Ibuprofen showed significant superiority over both aspirin and placebo, while aspirin did not exhibit significant superiority over placebo [24]. A recent study conducted in 2020 by Nie W, Xu P et al. compared the effectiveness and safety of naproxen, Ibuprofen, diclofenac, aspirin, and ketoprofen for treating primary dysmenorrhea. The study concluded that among these options, Ibuprofen was recommended as the preferred treatment for primary dysmenorrhea [10].

A novel finding of this study is that there was no significant difference in pain reduction between Ibuprofen (400 mg) and dark chocolate (70% cocoa, 35 g). This suggests that dark chocolate, with its high cocoa content, may possess pain-relieving properties comparable to Ibuprofen. Previous research has highlighted the potential benefits of dark chocolate in reducing pain, attributed to magnesium and its effects on prostaglandins. Magnesium helps to regulate the production of prostaglandins by inhibiting the enzyme that converts arachidonic acid into prostaglandins. Arachidonic acid is an essential fatty acid that is found in the body [25]. When arachidonic acid is converted into prostaglandins, it can cause inflammation and pain [26]. Further investigation is warranted to explore the mechanisms underlying the pain-relieving effects of dark chocolate and its potential as an alternative treatment for dysmenorrhea.

In contrast, both Ibuprofen and dark chocolate showed significant differences in pain reduction compared to young coconut water. The disparity in pain reduction may be attributed to the difference in magnesium content between the interventions. Magnesium has been recognized for its role in reducing menstrual pain intensity by inducing vasodilation in blood vessels and relaxing uterine muscles [27]. The significantly lower magnesium content in young coconut water (28 mg) compared to dark chocolate (59.5 mg) may explain the observed difference in effectiveness. Future research should explore the specific mechanisms by which magnesium contributes to pain reduction and further investigate the potential of magnesium-rich interventions in dysmenorrhea management.

This study has several limitations that should be acknowledged. First, blinding the participants was not feasible due to the differences in intervention characteristics, such as size, formulation, scent, and taste. Second, the study did not address treatment adherence because the data collectors were blinded to the intervention groups. This means that the data collectors did not know which participants were in which intervention group and, therefore, could not collect data on how well participants followed the treatment regimen. This could have affected the results, as participants who did not adhere to the treatment may have experienced different results than those who did. Third, the study only looked at the effects of a single dose of the interventions. It is possible that the effects of the interventions would be different if they were taken for multiple doses or over a longer period of time. Additionally, the study did not employ subject matching, which could have introduced confounding factors. Future studies should consider strategies to overcome these limitations, such as blinding techniques, more rigorous subject matching protocols, and longer treatment periods.

In conclusion, the current study provides evidence supporting the effectiveness of Ibuprofen in reducing menstrual pain intensity, as well as suggesting the potential pain-relieving properties of dark chocolate. The significant difference between these interventions and young coconut water highlights the importance of considering magnesium content in dysmenorrhea management. These findings contribute to our understanding of the comparative effectiveness of interventions and provide valuable insights for healthcare professionals and individuals seeking management options for primary dysmenorrhea. Further research is needed to explore the mechanisms underlying the pain-relieving effects of dark chocolate and the role of magnesium in dysmenorrhea management.

## 5. Conclusions

In conclusion, there are variations in the effectiveness of different intervention groups in reducing the intensity of primary dysmenorrhea. Specifically, Ibuprofen 400 mg was found to be the most effective intervention, while dark chocolate showed greater efficacy compared to young coconut water (330 mL). Although the difference in effectiveness between Ibuprofen 400 mg and dark chocolate (70% cocoa, 35 g) was not significant, there were notable disparities in effectiveness when comparing Ibuprofen with young coconut water, as well as dark chocolate with young coconut water.

To further enhance our understanding, it is recommended that future research should focus on investigating the efficacy of dark chocolate with higher cocoa percentages or in larger quantities. Additionally, considering that a significant proportion of individuals experiencing dysmenorrhea do not seek any form of treatment, further investigations into the underlying factors contributing to this trend would be valuable.

Practically, the findings of this study have implications for reproductive health education. Currently, the use of young coconut water and dark chocolate in dysmenorrhea management is underrepresented in educational materials, and comprehensive comparative data are lacking. It is, therefore, suggested that these findings enable healthcare professionals to offer alternative options, such as dark chocolate and young coconut water, as potential interventions for dysmenorrhea management.

Overall, this study sheds light on the varying effectiveness of interventions for primary dysmenorrhea and provides insights that can contribute to improved management strategies and reproductive health education.

## Figures and Tables

**Figure 1 ijerph-20-06619-f001:**
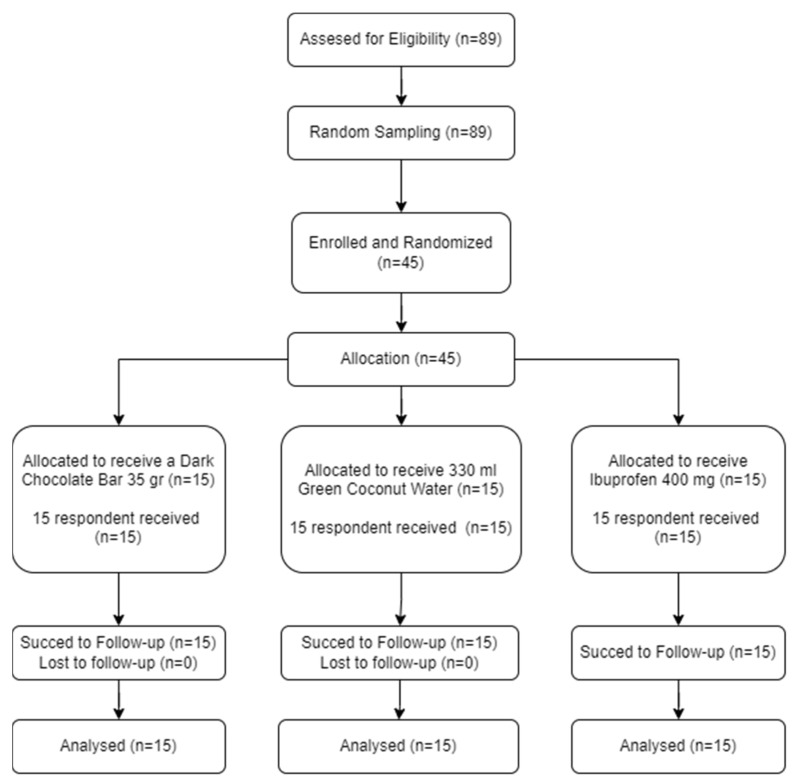
Flow chart of study participants.

**Table 1 ijerph-20-06619-t001:** The frequency distribution of subject characteristics.

Characteristics.	Intervention Group						Total		Chi-Square Test
	Dark Chocolate 70% 35 g		Young Coconut Water 330 mL		Ibuprofen 400 mg				
	f (n = 15)	%	f (n = 15)	%	f (n = 15)	%	f (n = 45)	%	*p*-Value
Age (Years)									
17 years	3	20	5	33.3	3	20	11	24.4	0.587
18 years	5	33.3	4	26.7	5	33.3	14	31.1	
19 years	2	13.3	4	26.7	2	13.3	8	17.8	
20 years	3	20	1	6.7	5	33.3	9	20	
21 years	2	13.3	1	6.7	0	0	3	6.7	
Body Mass Index (BMI)									
Underweight	1	6.7	0	0	0	0	1	2.2	0.583
Underweight (Mild)	0	0	1	6.7	0	0	1	2.2	
Normal	13	86.7	14	93.3	14	93.3	41	91.1	
Overweight (Mild)	1	6.7	0	0.0	1	6.7	2	4.4	
Menarche									
Early Menarche	1	6.7	2	13.3	0	0	3	6.7	0.653
Normal Menarche	12	80	12	80	13	86.7	37	82.2	
Late Menarche	2	13.3	1	6.7	2	13.3	5	11.1	
Menstrual Cycle									
Normal (28–35 days)	9	60	12	80	10	66.7	31	68.9	0.484
Oligomenorrhea (>35 days)	6	40	3	20	5	33.3	14	31.1	
Menstrual Duration									
>7 days	4	26.7	5	33.3	33	20	12	26.7	0.711
2–7 days	11	73.3	10	66.7	12	80	33	73.3	
Onset									
As soon as Menarche	15	100	15	100	15	100	45	100	-
Pain Management									
No Intervention	12	80	13	86.7	11	73.3	36	80	0.705
Warm Water Compress	2	13.3	2	13.3	2	13.3	6	13.3	
NSAIDs	1	6.7	0	0	2	13.3	3	6.7	
Impact of Pain on Daily Activities									
Tolerable	10	66.7	7	46.7	6	40	23	51.1	0.424
Partially Disruptive	5	33.3	6	40	8	53.3	19	42.2	
Unable to Perform Activities	0	0	2	13.2	1	6.7	3	6.7	

**Table 2 ijerph-20-06619-t002:** Distribution of pain intensity before and after treatment in the intervention group using crosstab analysis and chi-square test.

Pain Intensity	Intervention Group			Total	*p*-Value (Chi-Squaree-Test)
	Ibuprofen 400 mg	Coconut Water 330 mL	Dark Chocolate 70% 35 g		
**Pretest**					
Mild Pain (1–3)	1 (6.7%)	0 (0%)	1 (6.7%)	2 (4.4%)	0.411
Moderate Pain (4–6)	11 (73.3%)	11 (73.3%)	7 (46.7%)	29 (64.4%)	
Severe Pain (7–9)	3 (20%)	4(26.7%)	7 (46.7%)	14 (31.1%)	
**Posttest**					
No Pain (0)	10 (66.7%)	0 (0%)	5 (33.3%)	15 (33.3%)	0.004
Mild Pain (1–3)	4 (26.7%)	11 (73.3%)	7 (46.7%)	22 (48.9%)	
Moderate Pain (4–6)	1 (6.7%)	4(26.7%)	3 (20%)	8 (17.8%)	

**Table 3 ijerph-20-06619-t003:** Distribution of pain intensity before and after treatment in the intervention group.

Intervention Group	Kruskal Willis Test			
	n	NRS*	Mean Rank	*p*-Value
Ibuprofen 400 mg	15	4.6	31.23	0.000
Young Coconut Water 330 mL	15	2.87	12.97	
Dark Chocolate 70% 35 g	15	3.87	24.80	

*:.The average difference in the value of the Numeric Rating Scale before and after the intervention.

**Table 4 ijerph-20-06619-t004:** Post hoc test.

Intervention Group	Post-Hoc Test	
	*p*-Value	Significance
Ibuprofen 400 mg vs. Young Coconut Water 330 mL	0.000	Yes
Young Coconut Water 330 mL vs. Dark Chocolate Bar 70%	0.031	Yes
Dark Chocolate Bar 70% 35 g vs. Ibuprofen 400 mg	0.490	No

## Data Availability

Not applicable.

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
