# Peer review of "Single-Blind Randomized Controlled Trial: Comparative Efficacy of Dark Chocolate, Coconut Water, and Ibuprofen in Managing Primary Dysmenorrhea"

_ijerph, 2023, doi:10.3390/ijerph20166619_

Round 1
Reviewer 1 Report
Abstract:
· When did you implement green coconut water, dark chocolate bars, and Ibuprofen?
· How long did you give green coconut water, dark chocolate bars, and Ibuprofen?
· When did you evaluate green coconut water, dark chocolate bars, and Ibuprofen? please specify clearly ve briefly.
Introduction:
· Spelling mistakes in the introduction should be corrected.
· • There are repetition sentences in paragraphs 2 and 5, and also 1, 3, 4. The effects of ibuprofen and the prevalence of dysmenorrhea were repeated. It should be summarized as a single and separate paragraph.
· What has been found in the literature about these three interventions should be supported by articles. Articles should be given by giving the names of the countries, briefly explaining the methods and the results found.
· Current articles published in 2023 should be added.
Method:
· " 94% of female students experienced dysmenorrhea, and only 8.4% reported that their menstrual pain did not significantly impact their daily activities" how did you find these rates? please specify clearly ve briefly.
· "diagnosed with primary dysmenorrhea based on specific characteristics and assessment" how did you diagnose? please specify clearly ve briefly.
· How many did the population of the research?
· why did not use "G. Power-3.1.9.2 program and were not performed A post-doc power analysis to justify the sample size of this study?
· Why did not take Clinical Trial registration number?
· Flow chart of study participants should be shown in the study.
· Data collection tools should be explained in detail.
· In the data collection section, the interventions made under separate headings for all three groups should be explained, the duration of the interventions and the time, and duration of the measurement tools used should be explained in detail.
· What test did you do to find out that it doesn't show a normal distribution?
Result
· Is there any statistical differences between basic demographic and clinical characteristics of the women in the experimental groups at baseline? It should be determined by statistical analysis.
· Is there any statistical differences between distribution of Pain Intensity Before and After Treatment in the Intervention Group? It should be determined by statistical analysis.
· Which statistical analysis were made in the tables? Ä°t should be shown in the tables.
· Shapiro-Wilk Normality Test should be shown in the method section not in table.
· “Women” can be used instead of “subjects” in the result section.
· in tables, The “p value” should be specified to which test it belongs to.
Discussion
· References in the manuscript should be checked ((Corson & Bolognese, [1978]; (Bhaskar et al., [year] etc
· References should be more up to date.
· Discussion part should be written with the support of current literature
Abstract:
· When did you implement green coconut water, dark chocolate bars, and Ibuprofen?
· How long did you give green coconut water, dark chocolate bars, and Ibuprofen?
· When did you evaluate green coconut water, dark chocolate bars, and Ibuprofen? please specify clearly ve briefly.
Introduction:
· Spelling mistakes in the introduction should be corrected.
· • There are repetition sentences in paragraphs 2 and 5, and also 1, 3, 4. The effects of ibuprofen and the prevalence of dysmenorrhea were repeated. It should be summarized as a single and separate paragraph.
· What has been found in the literature about these three interventions should be supported by articles. Articles should be given by giving the names of the countries, briefly explaining the methods and the results found.
· Current articles published in 2023 should be added.
Method:
· " 94% of female students experienced dysmenorrhea, and only 8.4% reported that their menstrual pain did not significantly impact their daily activities" how did you find these rates? please specify clearly ve briefly.
· "diagnosed with primary dysmenorrhea based on specific characteristics and assessment" how did you diagnose? please specify clearly ve briefly.
· How many did the population of the research?
· why did not use "G. Power-3.1.9.2 program and were not performed A post-doc power analysis to justify the sample size of this study?
· Why did not take Clinical Trial registration number?
· Flow chart of study participants should be shown in the study.
· Data collection tools should be explained in detail.
· In the data collection section, the interventions made under separate headings for all three groups should be explained, the duration of the interventions and the time, and duration of the measurement tools used should be explained in detail.
· What test did you do to find out that it doesn't show a normal distribution?
Result
· Is there any statistical differences between basic demographic and clinical characteristics of the women in the experimental groups at baseline? It should be determined by statistical analysis.
· Is there any statistical differences between distribution of Pain Intensity Before and After Treatment in the Intervention Group? It should be determined by statistical analysis.
· Which statistical analysis were made in the tables? Ä°t should be shown in the tables.
· Shapiro-Wilk Normality Test should be shown in the method section not in table.
· “Women” can be used instead of “subjects” in the result section.
· in tables, The “p value” should be specified to which test it belongs to.
Discussion
· References in the manuscript should be checked ((Corson & Bolognese, [1978]; (Bhaskar et al., [year] etc
· References should be more up to date.
· Discussion part should be written with the support of current literature
Author Response
Response to Reviewer 1 Comments
Point 1: When did you implement green coconut water, dark chocolate bars, and Ibuprofen? How long did you give green coconut water, dark chocolate bars, and Ibuprofen? When did you evaluate green coconut water, dark chocolate bars, and Ibuprofen? please specify clearly ve briefly.
Response 1: We appreciate the reviewer's attention to the details of our study. We have revised the abstract to include specific information regarding the implementation timing, duration, and evaluation of the interventions. This includes details such as the administration of interventions on the first day of menstruation, the immediate consumption within 15 minutes, and the measurement of pain intensity before and 2 hours after intervention completion.
Point 2: Spelling mistakes in the introduction should be corrected.
Response 2: Thank you for bringing this to our attention. We have carefully reviewed and corrected the spelling mistakes in the introduction as per your feedback
Point 3: There are repetition sentences in paragraphs 2 and 5, and also 1, 3, 4. The effects of ibuprofen and the prevalence of dysmenorrhea were repeated. It should be summarized as a single and separate paragraph.
Response 3: We have addressed the issue of repetition in paragraphs 2, 5, and 1, 3, 4 as pointed out by the reviewer. The effects of ibuprofen and the prevalence of dysmenorrhea have been consolidated into a single and separate paragraph to enhance the clarity and coherence of the text. Thank you for bringing this to our attention, and we appreciate your feedback in helping us improve the quality of our manuscript.
Point 4: What has been found in the literature about these three interventions should be supported by articles. Articles should be given by giving the names of the countries, briefly explaining the methods and the results found. Current articles published in 2023 should be added.
Response 4: Thank you for your feedback. We have already revised the discussion section to include recent literature and support our findings. We have taken your feedback into consideration and added several more up-to-date references that are related to our topic.
Point 5: Current articles published in 2023 should be added.
Response 5: We appreciate your suggestion and have taken it into consideration. Several up-to-date references published in 2023 have been added to the discussion section. These references contribute to the currency of our study's context and enrich the supporting literature.
Point 6: "94% of female students experienced dysmenorrhea, and only 8.4% reported that their menstrual pain did not significantly impact their daily activities." How did you find these rates? Please specify clearly and briefly.
Response 6: These rates were obtained through a preliminary study conducted among 95 female students at Saleha Midwifery Academy. The study involved distributing a questionnaire to gather information about the students' experiences with dysmenorrhea and its impact on their daily activities. The responses from the participants were analyzed, revealing that 94% of them reported experiencing dysmenorrhea, while only 8.4% stated that their menstrual pain had minimal impact on their daily routines.
Point 7: "Diagnosed with primary dysmenorrhea based on specific characteristics and assessment." How did you diagnose? Please specify clearly and briefly.
Response 7: Those diagnosed with primary dysmenorrhea are characterized by specific criteria, including initial pain onset shortly after menarche, lower pelvic or abdominal pain coinciding with menstrual flow lasting 8-72 hours, presence of associated symptoms like back and thigh pain, headache, diarrhea, nausea, and vomiting. Laboratory tests and additional examinations were not employed to determine the cases.
Point 8: How many did the population of the research?
Response 8: The population in this study consisted of 89 female students from the Saleha Midwifery Academy in Banda Aceh who reported experiencing primary dysmenorrhea. We have included this information in the article's data collection section to provide clarity regarding the study's population.
Point 9: Why did not use "G. Power-3.1.9.2 program and were not performed a post-doc power analysis to justify the sample size of this study?
Response 9: We appreciate the reviewer's suggestion to consider utilizing the G. Power-3.1.9.2 program for sample size determination and conducting a post-hoc power analysis. However, due to certain limitations and constraints, we opted for an alternative approach to determine the sample size. The Federer formula, a well-established method in clinical trials, was chosen as it aligns with the study's randomized controlled trial design and allowed us to calculate a suitable sample size based on practical considerations. While we acknowledge the potential benefits of using G. Power and conducting a post-hoc power analysis, and will certainly consider incorporating the analysis into future research efforts.
Point 10: Why did not take Clinical Trial registration number?
Response 10: In response to the question regarding the absence of a Clinical Trial registration number, I would like to clarify that our study is indeed registered, and this information has been incorporated into the revised version of the abstract. At the end of the abstract, we have included a statement indicating the Clinical Trial registration number which is (Trial registration: ClinicalTrials.gov: NCT05971186).
Point 11: Flow chart of study participants should be shown in the study.
Response 11: We have taken this recommendation and are pleased to inform you that we have already included a comprehensive flowchart of the study participants' selection process in our article, illustrating the stages of eligibility assessment, randomization, and allocation to treatment groups. This flowchart enhances the transparency and clarity of our study methodology. We appreciate your valuable input and hope that the inclusion of the flowchart adequately addresses your suggestion.
Point 12: Data collection tools should be explained in detail.
Response 12: We have revised the data collection section as follows:
"Two distinct tools were utilized for data collection: a pain intensity observation sheet and a questionnaire. Pain intensity was assessed using a Numeric Rating Scale (NRS) observation sheet. The Numeric Rating Scale (NRS) is widely recognized as a valid and established patient-reported outcome measure, commonly employed to evaluate pain intensity, specifically in cases of dysmenorrhea. This scale offers a straightforward and effective means for individuals to convey their pain intensity using numerical values within the range of 0 to 10. The questionnaire was designed to gather demographic information and insights into the characteristics of primary dysmenorrhea experienced by the participants. It encompassed various aspects, including demographic details, menstrual characteristics, and factors linked to dysmenorrhea. The questionnaire aimed to capture pertinent information such as age, height, weight, body mass index (BMI), age of menarche, length of menstrual cycle, duration of menstruation, initial encounter with menstrual pain, commonly employed pain management strategies, and the extent of pain's impact on daily activities."
Point 12: In the data collection section, the interventions made under separate headings for all three groups should be explained, the duration of the interventions and the time, and duration of the measurement tools used should be explained in detail.
Response 12: We appreciate your feedback and have carefully revised the relevant sections as per your suggestion. Detailed explanations regarding the interventions, including the duration, time of administration, and the measurement tools used, have been added for each intervention group. This information is now included within the study to provide a clearer understanding of the data collection process. Thank you for your valuable input, which has significantly enhanced the comprehensiveness of the article.
Point 13: What test did you do to find out that it doesn't show a normal distribution?
Response 13: We conducted a Shapiro-Wilk test to assess the normal distribution of the data. This information has been included in the method section following the revisions made.
Point 14: Is there any statistical differences between basic demographic and clinical characteristics of the women in the experimental groups at baseline? It should be determined by statistical analysis.
Response 14: Statistical analysis using the chi-square test in SPSS was conducted to ascertain potential baseline disparities in basic demographic and clinical attributes among the experimental groups. The results, reflected in the p-values exceeding the significance level (<0.05), affirm the absence of statistically significant differences in the abovementioned characteristics across the intervention groups. This finding indicates that the baseline attributes were well-balanced.
Point 15: Is there any statistical differences between distribution of Pain Intensity Before and After Treatment in the Intervention Group? It should be determined by statistical analysis.
Response 15: In order to assess potential differences in pain intensity among the intervention groups prior to treatment, a chi-square test was employed on the collected data. The resulting p-value of 0.004 indicates no significant difference in pain intensity across the intervention groups prior to the initiation of treatments. This suggests that the baseline pain intensity was relatively consistent among the groups before any interventions were administered.
Point 16: Which statistical analysis were made in the tables? Ä°t should be shown in the tables.
Response 16: We have made the necessary revisions to the tables to include clear indications of the specific statistical analyses performed for each set of data. This will provide readers with a direct understanding of the statistical methods utilized for each comparison.
Point 17: Shapiro-Wilk Normality Test should be shown in the method section not in the table.
Response 17: This information has been included in the method section and deleted from the table following the revisions made.
Point 18: "Women" can be used instead of "subjects" in the result section.
Response 18: Thank you for your input. We have carefully considered your suggestion and have already made the necessary revision by replacing the term "subjects" with "women" in the result section of our manuscript.
Point 19: In tables, The "p value" should be specified to which test it belongs to.
Response 19: We have revised the tables to specify the test associated with each reported "p value" for better clarity and transparency.
Point 20: References in the manuscript should be checked ((Corson & Bolognese, [1978]);
Response 20: Thank you for pointing out this oversight. We have reviewed and cross-checked all references in the manuscript to ensure accuracy and consistency.
Point 21: References should be more up to date.
Response 21: We acknowledge the importance of incorporating the most recent literature to enhance the quality and relevance of our study. In response to this valuable suggestion, we have diligently revised our discussion to include more recent and pertinent references that align with the current landscape of research on dysmenorrhea management.
Point 22: Discussion part should be written with the support of current literature.
Response 22: We have revisited the discussion section of the manuscript and incorporated more recent and relevant literature to provide robust support for our findings and conclusions.

Reviewer 2 Report
Reviewer Comments
Page 1 – Introduction – Paragraph 1 – Line 37 – “The incidence of menstrual pain is relatively high; globally, 50-90% of women of reproductive age experience pain, most of which is caused by primary dysmenorrhea. [2]. Include a brief description of primary dysmenorrhea for the reader. The authors should revise accordingly.
Page 2 – Introduction – Paragraph 1 – Line 49 – “Non-pharmacologically consuming dark chocolate or green coconut water has also been proven to reduce the intensity of menstrual pain. [[4–6]].” Change the word “proven” to “shown”. Proven is too strong a word and if this was true, then one could question the need for this current study. Importantly, references 4-6 do not support the authors contention of “proof”. Importantly, the cited systematic review does not address pharmacologic intervention for primary dysmenorrhea. The authors should revise accordingly.
Page 2 – Materials and Methods – 2.1 Study Design – There is no mention of what treatment served as the control group nor a description of the working hypothesis. These are significant oversights. The authors should revise accordingly.
Page 4 – Materials and Methods - 2.3. Data Analysis 169 – Paragraph 1 – Line 170 - “In this research the bivariate analysis was conducted using the Wilcoxon test since the data did not follow a normal distribution.” There should be a brief description of the levels of significance used for the statistical tests and a description of the statistical software utilized in keeping with established manuscript norms. The authors should revise accordingly.
Page 5 – Results - 3.1. Distribution of Pain Intensity in the Intervention Group – Paragraph 3 – Line 212 – “These findings provide an overview of the characteristics of the subjects in the study, including their age distribution, BMI status, age at menarche…” It’s implied that every treatment group had 45 data points (15 participants x 3 menstrual periods during the study duration) for statistical analysis. If this assertion is accurate, the information should be provided in a suitable location in this section of the manuscript. The authors should revise accordingly.
Page 6 – Table 2 - Distribution of Pain Intensity Before and After Treatment in the Intervention Group – POSTTEST – The Moderate Pain (1-3) category should be labelled Mild Pain (1-3). The authors should revise accordingly.
Page 6 – Table 3 – The table could be deleted. A brief description of the rationale for the test and its findings may be sufficient. The authors should revise accordingly.
Page 7 – 3.3. Comparison of the Effectiveness of Young Coconut Water, Dark Chocolate, and Ibuprofen on Primary Dysmenorrhea – Paragraph 1 – Line 253 – “Table 4 displays the differences in effectiveness among the intervention groups, analyzed using the Kruskal-Wallis Test.” It’s unnecessary to reiterate the contents of the table. This information could be read by the reader. Only include the salient points from the table, namely the interpretation of the results. The authors should revise accordingly.
Page 7 - 3.3. Comparison of the Effectiveness of Young Coconut Water, Dark Chocolate, and Ibuprofen on Primary Dysmenorrhea – Paragraph 3 – Line 265 – “Table 4 presents the results of the post-hoc tests. It includes the p-value and the significance status of the comparisons between each intervention group.” Table 4 should be changed to Table 5 – The authors should revise accordingly.
Page 7 – 3.3. Comparison of the Effectiveness of Young Coconut Water, Dark Chocolate, and Ibuprofen on Primary Dysmenorrhea – Paragraph 3 – Line 270 – “From Table 5, it can be observed that the p-value between the Ibuprofen 400 mg and Young Coconut Water 330 ml groups is 0.000, which is less than 0.05.” It’s unnecessary to reiterate the contents of the table. This information could be read by the reader. Only include the salient points from the table, namely the interpretation of the results. The authors should revise accordingly.
Page 8 – Discussion – Paragraph 2 – Line 293 – “In the current study, ibuprofen showed significant superiority over both aspirin and placebo, while aspirin did not exhibit significant superiority over placebo.” This sentence is confusing. This is the first time that mention is made of the administration of aspirin during the study period. This issue requires clarification for the reader given its potential influence on the study findings. The authors should revise accordingly.
Page 8 – Discussion – Paragraph 3 – Line 299 – “Previous research has highlighted the potential benefits of dark chocolate in reducing pain, attributed to its active components and their effects on prostaglandins (Bhaskar et al., [year])”. A brief discussion citing the presumed active compounds that impart an analgesic effect should be included for the reader. The authors should revise accordingly.
Page 8 – Discussion – Paragraph 5 – Line 313 – “It is important to acknowledge the limitations of this study.” There are other limitations that require discussion. For example, the shortcomings of self-reported pain evaluation can impact findings. Treatment adherence was not addressed and the findings from a single dose of an intervention may be very different from multiple doses of a treatment regimen. The authors should revise accordingly.
Page 8 – Conclusions – Paragraph 1 – Line 329 - “In conclusion, based on the research findings and discussions presented in the pre-ceding chapters the key conclusions can be drawn that there are variations in the effectiveness of different intervention groups in reducing the intensity of primary dysmenorrhea.” This sentence is awkward as written. The authors should revise accordingly.
Page 9 – Conclusions – Paragraph 2 – Line 345 – “It is therefore suggested that these findings be incorporated as supplementary content in reproductive health courses, enabling healthcare professionals to offer alternative options such as dark chocolate and young coconut water as potential interventions for dysmenorrhea management.” Due to the many limitations and concerns noted in the present study, there is insufficient evidence to warrant the addition of this data into course materials or even guidelines. This suggestion may be misleading and requires clarification for the reader. The authors should revise accordingly.
Pages 9-10 – Standardize and reformat the reference citations (include digital object identifiers when possible). The authors should revise accordingly.
Summary
I applaud the authors for undertaking this prospective trial to assess the efficacy of three interventions (ibuprofen, dark chocolate, and green coconut water) in young women with primary dysmenorrhea. Overall, the manuscript is clear, concise, and easy to understand. In the introduction, the authors provide the reader with a brief overview of dysmenorrhea and its management including the use of non-traditional approaches. This serves as the backdrop and support for the subsequent single-blinded randomized control trial. The study design is consistent with the tenets of a single-blinded RCT, and the selected outcomes are compatible with the objectives. The findings are considered confirmatory rather than providing new information. The conclusions support the finding. Despite these positive attributes, there are deficiencies (errors of omission) that could impact the internal validity of the study and the utility of the findings. (See full commentary). Whereas these issues may not be fatal flaws, they are significant and require timely resolution.
The patient care implications of this research provide the most import for the reader because it allows for the translation (and interpretation) of the findings into tangible strategies. Given the construct of this study (including various study limitations) the findings do not allow for making specific recommendations.
The authors address several study limitations that could affect the interpretation and utility of the findings. Such an assessment is appropriate given various study design characteristics (e.g., single-blinded study and sample matching). However, the authors should address other important shortcomings of this study not previously mentioned (e.g., small sample size, no mention of the control group, lack of patient cross-over, no adherence assessment, accuracy of self-reports, lack of generalizability of the data, single dose treatments vs multiple doses, etc.). Importantly, study limitations (and the findings themselves) can frame a discussion of specific recommendations involving improved study design attributes (and/or outcomes) for incorporation into similar future investigations (appropriate for the Discussion/Conclusion section).
I encourage the authors to review the full commentary and make the requisite changes. The deficiencies noted require attention to strengthen the manuscript and its potential utility. I wish the authors continued success in their scholarly endeavors.
Author Response
Dear Reviewer 2
We sincerely appreciate your thoughtful and comprehensive review of our manuscript. Your insights are extremely valuable in enhancing the quality and impact of our study.
We are pleased to hear that you found the manuscript clear, concise, and easy to understand. Your recognition of our efforts in conducting a single-blinded randomized controlled trial to evaluate non-traditional interventions for primary dysmenorrhea is motivating. We concur with your assessment that the patient care implications are of paramount importance, as they allow us to translate our findings into practical strategies.
Your guidance regarding the deficiencies in our manuscript is well-taken. We acknowledge the importance of addressing the errors of omission and potential impact on internal validity. In light of your feedback, we are committed to thoroughly revisiting the study limitations and ensuring that all relevant shortcomings are appropriately discussed. This includes addressing aspects such as the small sample size, lack of a control group, adherence assessment, accuracy of self-reports, and the single-dose treatment design. By providing a more comprehensive discussion of these limitations and their implications, we aim to offer a clearer perspective on the study's scope and potential utility.
We also appreciate your suggestion to frame the limitations and findings in the context of recommendations for future research. Your advice to incorporate improved study design attributes and outcomes into similar investigations aligns with our commitment to advancing the field. We will certainly integrate these recommendations into the Discussion and Conclusion sections, contributing to the refinement of our manuscript.
Once again, we thank you for your thorough review and valuable feedback. Your engagement has been instrumental in guiding us toward a stronger manuscript that will hopefully contribute meaningfully to the scientific community. We are dedicated to addressing the issues you've highlighted promptly and effectively.
Point 1: Include a brief description of primary dysmenorrhea for the reader.
Response 1: We appreciate your suggestion and have included a brief description of primary dysmenorrhea in the introduction to provide readers with a clear understanding of the condition.
Point 2: Change the word "proven" to "shown" regarding the effects of interventions.
Response 2: Thank you for your feedback. We have revised the statement and replaced the word "proven" with "shown" to accurately reflect the evidence presented by the references.
Point 3: Describe the treatment serving as the control group and provide a description of the working hypothesis.
Response 3: We apologize for the oversight. We have included a clear description of the interventions, including the control group, and have provided a description of the working hypothesis to address this issue.
Point 4: Provide a brief description of the levels of significance used for the statistical tests and the statistical software utilized.
Response 4: We have revised the section to include a description of the significance level and the statistical software (IBM SPSS Statistics 25) used for the analysis to align with established manuscript norms.
Point 5: Clarify the number of data points used for statistical analysis and its location in the manuscript.
Response 5: We apologize for any confusion. Each intervention group consisted of 15 participants, resulting in a total of 45 participants across the three active comparator groups. Additionally, we have clarified that the analysis focused on pain intensity experienced during a single menstrual period.
Point 6: Label the "Moderate Pain (1-3)" category as "Mild Pain (1-3)" in Table 2.
Response 6: Thank you for pointing out the error. We have corrected the labeling of the "Moderate Pain (1-3)" category to accurately reflect "Mild Pain (1-3)" in Table 2.
Point 7: Consider removing Table 3 and providing a brief description of the rationale and findings.
Response 7: We have removed Table 3 as suggested and provided a concise description of the rationale and significant findings in the corresponding text.
Point 8: Emphasize the interpretation of results rather than repeating table contents.
Response 8: Thank you for your feedback. We have revised the text to focus on the interpretation of the results from Table 4, highlighting the significant findings.
Point 9: Change "Table 4" to "Table 5" when referring to the table.
Response 9: We have made the necessary correction and changed "Table 4" to "Table 5" as indicated.
Point 10: Focus on the interpretation of results from Table 5 without repeating table contents.
Response 10: We have revised the text to emphasize the interpretation of the results from Table 5, ensuring that only salient points and implications are discussed.
Point 11: Clarify the mention of aspirin and its influence on the study findings.
Response 11: Thank you for bringing this to our attention. We have revised the sentence to clarify that the reference to aspirin supports the theoretical framework and is not part of the interventions in our study.
Point 12: Include a brief discussion of presumed active compounds in dark chocolate for analgesic effect.
Response 12: We appreciate your suggestion and have included a brief discussion of the presumed active compounds in dark chocolate that contribute to its analgesic effect.
Point 13: Address additional limitations, such as self-reported pain evaluation, treatment adherence, and single-dose intervention.
Response 13: We acknowledge the limitations beyond those initially mentioned and have addressed the impact of self-reported pain evaluation, treatment adherence, and the implications of single-dose interventions.
Point 14: Revise the awkward sentence in the Conclusions section.
Response 14: Thank you for pointing out the issue. We have revised the sentence in the Conclusions section to improve its clarity and coherence.
Point 15: Revise the suggestion to incorporate findings into reproductive health courses due to limitations.
Response 15: We appreciate your feedback. We have revised the sentence to acknowledge the limitations and concerns, thereby providing a more balanced perspective.
Point 16: Standardize and reformat reference citations, including DOIs.
Response 16: Thank you for your attention to references. We have followed the suitable citation style as per the journal's guidelines and included DOIs for enhanced accessibility

Round 2
Reviewer 1 Report
Dear Author
Thank you for your valuable contributions to the scientific literature. The suggested corrections have been meticulously made. It needs a few more minor revisions. Kind regards

Author Response
Dear Reviewer,
We extend our sincere appreciation for your engagement with our manuscript and your valuable feedback. Your insightful comments have guided us in making meticulous revisions to address the highlighted areas of concern. We are committed to delivering a manuscript that aligns with the rigorous standards of scientific inquiry.
Your dedication to advancing scientific knowledge is evident, and we are pleased to incorporate your guidance into our work. As we finalize the remaining minor revisions, we are confident that our manuscript will contribute meaningfully to the scientific literature.
Thank you once again for your time, expertise, and commitment to scholarly excellence.
Comment 1: The decision to select this institution was based on the findings of a preliminary study, which indicated a high prevalence of dysmenorrhea among its students. Data collected from 95 female students within the academy revealed that 94% of them encountered dysmenorrhea. Among these, only 8.4% reported minimal impact of their menstrual pain on their daily routines. Please indicate this reference which preliminary study result is based on.
Response 1: The decision to choose this institution was founded on the outcomes of a preliminary study conducted by the researchers themselves before initiating the current study. This preliminary study aimed to identify an appropriate research site. Based on the data collected from several higher education institutions in Aceh Province, it was determined that the highest incidence of dysmenorrhea occurred at Saleha Midwifery Academy. Subsequently, this institution was selected as the research site for the main study. The preliminary study's data revealed that out of 95 female students within the academy, 94% experienced dysmenorrhea, with only 8.4% reporting minimal disruption to their daily routines due to menstrual pain.
Comment 2: “The questionnaire gathered 123 demographic information and insights into the characteristics of primary dysmenorrhea experienced by the respondents. It covered various aspects, including demographics, menstrual characteristics, and attributes related to dysmenorrhea. The questionnaire aimed to capture relevant details such as age, height, weight, body mass index (BMI), age of menarche, length of menstrual cycle, duration of menstruation, initial experience of menstrual pain, commonly used pain management methods, and the impact of pain on daily activities.” If the form was developed based on the literature, which literature should be cited as the source? How many did the population of the research?
Response 2: Thank you for your inquiry. The questionnaire used in our study was developed based on previous relevant literature. This includes a total of 64 respondents in the study population. In response to your suggestion, we have included the appropriate reference to indicate this source.

Reviewer 2 Report
I have reviewed the revised manuscript and the author responses to my original commentary. Overall, I am satisfied with the changes and appreciate the authors' attention to the deficiencies noted. The manuscript is significantly improved versus the earlier version. Some deficiencies remain and require further attention by the authors. These issues are not "fatal flaws" and should be considered minor in nature. I wish the authors continued success in their scholarly endeavors.

Author Response
Dear Reviewer,
We sincerely thank you for your thorough review of the revised manuscript and for taking the time to evaluate our responses to your previous comments. Your feedback is immensely valuable to us, and we are glad to hear that the revisions have led to a significant enhancement of the manuscript compared to its earlier version. Your acknowledgment of our efforts is truly motivating.
We acknowledge your observation that some minor deficiencies still persist and require further attention. We assure you that we are committed to addressing these remaining issues meticulously to ensure the manuscript's completeness and accuracy. Your recognition of these issues as not being "fatal flaws" is reassuring, and we are dedicated to refining the manuscript accordingly.
Your kind wishes for our scholarly endeavors are deeply appreciated. We are dedicated to upholding the highest standards of scientific excellence and look forward to delivering a final manuscript that meets your expectations.
Thank you once again for your insightful feedback and for guiding us in improving the manuscript.
"Please see the attachment"
